# Effect of Carotenoids from *Phaeodactylum tricornutum* on Palmitate-Treated HepG2 Cells

**DOI:** 10.3390/molecules25122845

**Published:** 2020-06-19

**Authors:** Claire Mayer, Martine Côme, Vincent Blanckaert, Graziella Chini Zittelli, Cecilia Faraloni, Hassan Nazih, Khadija Ouguerram, Virginie Mimouni, Benoît Chénais

**Affiliations:** 1EA 2160 MMS, Mer Molécules Santé, IUML FR 3473 CNRS, Institut Universitaire Technologique, Le Mans Université, F-53020 Laval CEDEX 9, France; claire.mayer@univ-lemans.fr (C.M.); martine.come@univ-lemans.fr (M.C.); vincent.blanckaert@univ-lemans.fr (V.B.); virginie.mimouni@univ-lemans.fr (V.M.); 2National Research Council, Department of Biology, Agriculture and Food Sciences, Institute for BioEconomy, I-50019 Sesto Fiorentino (Florence), Italy; graziella.chinizittelli@cnr.it (G.C.Z.); faraloni@cnr.it (C.F.); 3EA 2160 MMS, Mer Molécules Santé, IUML FR 3473 CNRS, UFR Pharmacie, Université de Nantes, F-44035 Nantes CEDEX 1, France; el-hassane.nazih@univ-nantes.fr; 4UMR1280 PhAN, Physiopathology of Nutritional Adaptations, INRAe, University of Nantes, CHU Hôtel Dieu, IMAD, CRNH Ouest, F-44000 Nantes, France; khadija.ouguerram@univ-nantes.fr; 5EA 2160 MMS, Mer Molécules Santé, IUML FR 3473 CNRS, UFR Sciences et Techniques, Le Mans Université, F-72085 Le Mans CEDEX 9, France

**Keywords:** *Phaeodactylum tricornutum*, total lipophilic extract, carotenoid extract, *n*-3 LC-PUFA, lipid metabolism, gene expression, HepG2 cells, palmitate, non-alcoholic fatty liver disease

## Abstract

Non-alcoholic fatty liver disease represents the most common liver disease and is characterized by an excess of lipid accumulation in hepatocytes, mainly stored as triglycerides. *Phaeodactylum tricornutum* is a marine microalga, which is rich in bioactive molecules known to be hepatoprotective, such as *n*-3 long-chain polyunsaturated fatty acids and fucoxanthin. The aim of this study was to investigate the effects of a carotenoid extract from *P. tricornutum* in a cellular model of non-alcoholic fatty liver disease induced by palmitate treatment. The combined effects of carotenoids and lipids, especially *n*-3 long-chain polyunsaturated fatty acids, were also investigated by using a total lipophilic extract. HepG2 cells were exposed for 24 h to 250 µM palmitate with or without the addition of carotenoid extract (6 μg/mL) or total lipophilic extract (100 μg/mL). The addition of carotenoid extract or total lipophilic extract prevented the accumulation of triglycerides, total cholesterol and cholesterol esters. The carotenoid extract and total lipophilic extract also decreased the mRNA expression levels of genes involved in lipogenesis (*ACACA*, *FASN*, *SCD* and *DGAT1*) and cholesterol esterification (*ACAT1*/*SOAT1*). In addition, the total lipophilic extract also downregulated the *LXR/NR1H3* and *SREBF1* genes, which are involved in lipogenesis regulation. By contrast, the carotenoid extract increased the mRNA level of *CPT1A*, a β-oxidation related gene, and reduced the lipid droplet accumulation. In conclusion, this study highlights the preventive effects against non-alcoholic fatty liver disease of the two microalga extracts.

## 1. Introduction

The changing lifestyle of our society, marked by an over-consumption of saturated fatty acids and sugars and a decrease in physical activity, is the main cause of the increase in the number of people with non-alcoholic fatty liver disease (NAFLD) [1,2]. NAFLD is currently the most common chronic liver disease in Western countries and affects more than 30% of the general population [2,3]. NAFLD is characterized by an excessive accumulation of lipids in the parenchymal cells of the liver in the absence of excessive alcohol consumption [3]. Based on the analyses of the Dallas Heart Study cohort, NAFLD was determined by a triglyceride level above 5.5% in the liver [4]. NAFLD is a consequence of obesity and metabolic syndrome [3,4,5]. Moreover, NAFLD can progress to a more serious stage, namely non-alcoholic steatohepatitis, which can lead to hepatic fibrosis [3].

De novo lipogenesis plays a major role in fatty acid metabolism and, in particular, contributes significantly to triglyceride accumulation in the hepatocytes of NAFLD patients [6]. Acetyl-CoA carboxylase (ACACA) converts acetyl-CoA to malonyl CoA, and diacylglycerol o-acyltransferase (DGAT) enzymes use acyl-CoA as a substrate as well as diacylglycerols to form triglycerides [6,7]. During lipid droplet formation, in addition to triglycerides, cholesterol esters are generated by the enzymes acyl-CoA cholesterol acyltransferases (ACATs), also called sterol o-acyltransferase (SOAT) [7]. The stearyl-CoA desaturase (SCD) is also an enzyme of de novo lipogenesis that catalyzes the synthesis of monounsaturated fatty acids from saturated fatty acids and that is involved in the development of insulin resistance [8]. In NAFLD patients, an increase in SCD activity was observed in the liver [9]. The de novo synthesis of triglycerides is promoted by the activation of the sterol regulatory element-binding transcription factor 1 (SREBF1), which is a target of nuclear liver X receptor (LXR/NR1H3) [10,11]. In order to limit the accumulation of fatty acids, liver cells increase mitochondrial β-oxidation by increasing the input of fatty acids through carnitine palmitoyltransferase-1 (CPT1) and CPT2 [12]. However, an increase in mitochondrial β-oxidation decreases the antioxidant capacity and causes increased oxidative stress [12].

Several reports have highlighted that *n*-3 long-chain polyunsaturated fatty acids (*n*-3 LC-PUFA), particularly eicosapentaenoic acid (EPA) and docosahexaenoic acid (DHA), reduce NAFLD progression by decreasing hepatic contents in triglycerides and cholesterol esters [13,14]. These beneficial effects could be explained by the ability of these *n*-3 LC-PUFA to control the activity and/or expression of transcription factors that regulate the expression of genes encoding proteins involved in de novo lipogenesis, fatty acid oxidation, fat uptake from the circulation and its assimilation into lipids, and very low density lipoprotein assembly and secretion [15]. Moreover, it was demonstrated that *n*-3 LC-PUFA exert preventive effects against dyslipidemia, inflammation, insulin-resistance, oxidative stress, and metabolic disturbances, all involved in the progression and establishment of NAFLD [16,17,18,19,20,21]. *n*-3 LC-PUFA are abundantly present in fish such as salmon, herring or cod and are mainly marketed as oil [22]. However, the decrease in fishery resources linked to marine pollution necessitates the search for new alternative sources of *n*-3 LC-PUFA [23].

Investigations of the potential effects of biomolecules from marine sources other than fish oil on NAFLD are crucial for new prevention strategies. To remedy this problem, *n*-3 LC-PUFA derived from microalgae could represent an interesting alternative. Indeed, thanks to their first place in the food chain, microalgae are less sensitive to heavy metal contamination [24]. Moreover, microalgae are a potential source of highly bioactive secondary metabolites such as pigments, sterols and soluble fibers and offer better nutritional value than fish oils [25]. Pigments, especially fucoxanthin, have also shown preventive effects against NAFLD, thanks to their high anti-obesogenic, anti-dyslipidemic, antioxidant and anti-inflammatory activities [26,27,28,29].

*P. tricornutum* is a ubiquitous marine diatom, located particularly in coastal areas subjected to salinity variations [30]. *P. tricornutum* is generally found in temperate or cold waters and can be pelagic or benthic [30]. Its appeal lies in its high level of EPA and fucoxanthin [31,32,33]. Some studies conducted in rodents have shown the hepatoprotective effects of microalgal lipophilic extracts rich in *n*-3 LC-PUFA against NAFLD [34,35]. However, to the best of our knowledge, no in vitro studies have demonstrated the protective effect of a pigment extract of *P. tricornutum* against NAFLD, whereas a recent study showed a reduction in fat accumulation and liver triglycerides in C57BL/6J mice supplemented with a *P. tricornutum* extract rich in fucoxanthin [36]. In addition, in a previous study in Wistar rats fed a high-fat diet supplemented with *P. tricornutum* at a dose of 12%, we also observed preventive effects with respect to NAFLD [37].

Thus, the main objective of this work was to study the efficiency of carotenoids from *p. tricornutum*, combined or not with lipids, in preventing the cellular features of NAFLD. The retained cellular model was the human hepatocarcinoma HepG2 cell line, which is commonly used to study lipid metabolism and metabolic disturbances in the liver, particularly those associated with NAFLD [38,39]. NAFLD was simulated by the treatment of cells with 250 µM palmitate for 24 h, as commonly used to induce lipotoxicity and hepatic steatosis in HepG2 cells [40,41,42].

## 2. Results

### 2.1. Effects of Carotenoid and Total Lipophilic Extracts on Cytotoxicity at 24 h of Treatment

The cytotoxicity of *P. tricornutum* extracts towards HepG2 cells was assayed after 24 h of treatment with increasing amounts of carotenoid extract (CE) and total lipophilic extract (TLE) (Figure 1a). The results showed the absence of significant cytotoxicity of the TLE for all the tested concentrations, ranging from 6 to 100 µg/mL, according to both the 3-(4,5-dimethylthiazol-2-yl)-2,5-diphenyltetrazolium bromide (MTT) assay (Figure 1a) and the lactate dehydrogenase (LDH) activity assay (Figure 1b). By contrast, CE induced an increase in LDH activity at the higher concentrations tested, i.e., 50–100 µg/mL (Figure 1a), and a decrease in cell viability at 75 µg/mL (Figure 1b). Therefore, further experiments were using 6 µg/mL of CE and 100 µg/mL of TLE.

### 2.2. Carotenoid Extract Decreased the Palmitate (PAL)-Induced Lipid Droplet Accumulation

Lipid droplet accumulation was estimated by Oil Red staining (Figure 2a) and quantified after the solubilization of lipid droplets (Figure 2b), showing that PAL treatment increased lipid accumulation in the control cells. The addition of CE, which had no significant effect in control cells, prevented the PAL-induced lipid droplet accumulation (Figure 2b). By contrast, adding TLE induced an increase in the lipid droplet accumulation (Figure 2b). Moreover, the addition of TLE did not prevent the PAL-induced lipid droplet accumulation (Figure 2b).

### 2.3. Carotenoid and Total Lipophilic Extracts Decreased Cellular Levels of Triglycerides, Total Cholesterol and Cholesterol Esters

As expected, and in correlation with the above results, the PAL treatment of HepG2 cells resulted in increased cellular levels of triglycerides, total cholesterol and cholesterol esters (Figure 2c–e). The addition of either CE or TLE to control cells had no significant impact on the lipid contents, with the exception of the cholesterol ester level, which increased in the presence of CE (Figure 2e). However, both CE and TLE prevented the PAL-induced increase in triglycerides, total cholesterol and cholesterol esters from baseline levels in HepG2 cells (Figure 2c–e).

### 2.4. Impact of Carotenoid and Total Lipophilic Extracts from P. tricornutum on mRNA Levels of Lipid Metabolism and β-Oxidation Related Genes

The mRNA levels for some lipogenic enzyme genes (i.e., *ACACA*, *DGAT1* and *FASN*) were not affected by the PAL treatment (Figure 3a). By contrast, the mRNA levels of the *SCD*, *ACAT1*/*SOAT1* and *SREBF1* genes were increased in PAL-treated cells compared to control (CTRL) cells (Figure 3a). The addition of CE or TLE to PAL-treated cells induced a decrease in the mRNA levels of lipogenic genes such as *ACACA*, *DGAT1* and *FASN* (Figure 3a). In the same way, the mRNA level of *LXR/NR1H3* was decreased in PAL-treated cells by microalgae extracts and more markedly by TLE (Figure 3a). Moreover, CE and TLE restored the mRNA level of the *ACAT1*/*SOAT1* gene in PAL-treated HepG2 cells, and only PAL + TLE treatment restored the mRNA level of *SREBF1* (Figure 3a). Both CE and TLE prevented the PAL-induced increase in the mRNA level of SCD above the baseline level, and even a decrease was seen with CE (Figure 3a). It should be noted that CE and TLE had an impact on CTRL cells, decreasing the *DGAT1* and *FASN* mRNA levels, and only TLE decreased the *ACACA* mRNA level in CTRL cells (Figure 3b). In CTRL cells, the *LXR/NR1H3* mRNA level was decreased by TLE and, conversely, increased by CE (Figure 3b).

The mRNA levels of genes related to fatty acid catabolism were also studied and may be increased in PAL-treated cells, as suggested by the increase in *CPT1A* and *CPT2* mRNA levels (Figure 3a). The addition of CE strongly increased the mRNA level of *CPT1A* in PAL-treated cells, while TLE had no effect. However, both CE and TLE decreased the mRNA level of *CPT2* below the basal level in CTRL and PAL-treated cells (Figure 3a,b).

## 3. Discussion

The aim of this study was to investigate the impact of CE from *P. tricornutum* on NAFLD and associated metabolic pathways, in human liver HepG2 cells. Twenty-four hours of PAL treatment at a concentration of 250 μM is well known to induce NAFLD without altering cell viability in HepG2 cells [40,41,42,43,44]. To assess the protective effects of microalga extracts on PAL-induced NAFLD, extracts have to be used at non-toxic concentrations. Here, CE showed some cytotoxic effects at high concentrations, increasing cell necrosis at concentrations between 50 µg/mL and 100 µg/mL. Accordingly, a recent study reported a decrease in cell metabolic activity associated with an increase in apoptosis in HepG2 cells treated with 50 µg/mL of fucoxanthin from *P. tricornutum* [45]. By contrast, TLE showed no toxicity toward HepG2 cells, suggesting that the presence of lipids and/or chlorophylls (chlorophyll a and derivatives such as pheophytin) counteracts the cytotoxicity of carotenoids. Indeed, according to the literature, chlorophylls have a chelating activity toward reactive oxygen species, which is used for pharmaceutical benefits, especially in liver recovery, and *n*-3 LC-PUFA have hepatoprotective effects, in particular through the boosting of the antioxidant system [46,47].

In agreement with several studies [40,41,42], PAL-treated HepG2 cells showed a significant increase in lipid droplet accumulation and the cellular levels of triglycerides, total cholesterol and cholesterol esters. Both extracts prevented the elevation of triglyceride, total cholesterol and cholesterol ester levels in HepG2 cells. However, only CE reduced the lipid droplet accumulation induced by PAL treatment. These results suggest that microalga extracts may have the potential to prevent NAFLD. These biological effects could be attributed to fucoxanthin, which is the main bioactive component of CE (Appendix A), or fucoxanthin and EPA, which are present in the TLE (Appendix A). Accordingly, EPA was reported to have a preventive effect on lipid accumulation in HepG2 cells co-treated with PAL and EPA (50 µM) for 24 h, which could be mediated by a decrease in hepatic fatty acid synthesis [48]. In addition, Wan et al. have shown in a rodent study using a lipid extract from *Chlorella pyrenoidosa*, which is relatively rich in EPA (~2% dw) and DHA (~4% dw), at doses of 150 and 300 mg/kg of body weight that lipid extract supplementation prevented NAFLD by reducing the lipid contents in the livers of rats fed high-fat and high-sucrose diets for 8 weeks [33]. A preventive effect against NAFLD has also been shown for fucoxanthin, originating from *P. tricornutum* (23.54 μg/mg of fucoxanthin by dry weight), which caused a reduction in fat accumulation and liver triglycerides in C57BL/6J mice [36]. Moreover, in our study, only the CE decreased the lipid droplet accumulation. Accordingly, fucoxanthin has previously been reported to inhibit lipogenic liver enzymes such as glucose-6-phosphate dehydrogenase, malic enzyme, FASN and phosphatidate phosphohydrolase, all involved in lipid droplet formation [27]. Besides, lipid droplet accumulation was not significantly affected in HepG2 cells treated with PAL + TLE, probably due to the direct addition of lipids, since *P. tricornutum* is well known to be rich in neutral lipids (about 30% of the lipid fraction) [49].

The de novo synthesis of fatty acids and fatty acid oxidation in the mitochondria are crucial steps of fatty acid metabolism in the liver [50]. The modulation of the gene expression of enzymes involved in these pathways could explain the preventive effect of microalga extracts against the NAFLD induced by PAL treatment. Firstly, PAL is known to increase lipid storage in the liver by a process mainly controlled by the SREBF1 transcription factor [51]. In accordance with the present study, previous studies using HepG2 cells treated for 24 h with palmitate at 250 µM or 300 µM showed an increase in the expression of enzymes involved in the oxidation and de novo synthesis of fatty acids such as CPT1A and SCD [41,52]. Although our analyses showed no significant change in the mRNA levels of lipogenic genes (*ACACA*, *DGAT1*, *FASN* and *LXR/NR1H3*) in our cellular model of NAFLD, the mRNA levels of *CPT2* and *ACAT1*/*SOAT1* were increased in agreement with in vivo studies [53,54].

In the present study, the decrease in the cellular triglyceride level in the presence of CE or TLE in PAL-treated cells is correlated with a decrease in the mRNA levels of lipogenic enzyme genes such as *ACACA*, *FASN*, *DGAT1* and *SCD*. In agreement with the literature, the fucoxanthin contained in both CE and TLE could regulate the gene expression of enzymes involved in lipid metabolism such as ACACA, FASN, DGAT1 and SCD [27,55,56,57,58]. Moreover, *n*-3 LC-PUFA (mainly EPA), which are present in TLE, have been reported to inhibit the hepatic activity of DGAT and SCD [59,60].

In addition, in cells co-treated with PAL and TLE, the decrease in the mRNA levels of lipogenic enzymes could be mediated upstream by the downregulation of the transcription factor SREBF1 [20]. The decrease in the *SREBF1* mRNA level observed here in the presence of PAL + TLE could be mediated by the decrease in the *LXR/NR1H3* transcription factor mRNA level. Indeed, *n*-3 LC-PUFA, especially EPA from the TLE, have already shown to have an inhibitory effect on the mRNA levels of *LXR*/*NR1H3* and genes involved in fatty acid synthesis such as *FASN* and *ACACA* [48,61]. Furthermore, *n*-3 LC-PUFA also have an inducing effect on the gene expression of the *farnesoid X receptor* transcription factor, leading to a decrease in the *SREBF1* mRNA level [61]. Although the mRNA level of *LXR/NR1H3* was decreased in PAL + CE cells, the *SREBF1* mRNA level was not reduced. These observations suggest that CE could act directly on the decrease in the mRNA level of lipogenic genes and not by inhibiting the expression of the *SREBF1* gene.

The decrease in the cellular levels of total and esterified cholesterol (a major form of cholesterol in the lipid droplets) was observed in the presence of CE or TLE in PAL-treated cells and could be correlated with the decrease in the *ACAT1*/*SOAT1* mRNA level, a gene involved in the esterification of cholesterol [7]. Moreover, the fucoxanthin present in both microalga extracts could decrease the mRNA level of *ACAT1*/*SOAT1* in HepG2 cells [27].

Finally, only PAL + CE treatment increased the mRNA level of *CPT1A*, suggesting the ability of the fucoxanthin from CE to decrease lipid droplet accumulation and lipid synthesis, via the activation of the β-oxidation pathway [62,63,64]. Surprisingly, the *CPT1A* mRNA level in PAL + TLE cells was unchanged compared to that in cells treated with PAL, suggesting that complex interactions between fucoxanthin and lipids from TLE may counteract the promoting effects of fucoxanthin and *n*-3 LC-PUFA on the β-oxidation pathway [27,35,62,63,64,65]. Thus, CE appears to have modulating effects on lipid metabolism through decreasing the mRNA levels of lipogenesis genes as well as increasing the mRNA level of *CPT1A*, involved in fatty acid oxidation, suggesting that CE may be an effective anti-NAFLD agent. The comparison of CE and TLE also highlights the combined action of lipids (mainly EPA) and pigments (carotenoid and chlorophylls), showing complex interactions.

## 4. Materials and Methods

### 4.1. Microalgal Extracts and Palmitate Solution

The TLE and CE from the microalga *P. tricornutum* were obtained from the CNR-IBE laboratory (Florence, Italy). TLE was prepared from freeze-dried biomass using three rounds of chloroform/methanol (2:1, *v*/*v*) extraction as described by Ma et al. [66] with slight modifications, and the dry extract was finally dissolved in 95% ethanol. Carotenoids were extracted from the biomass using 90% acetone, and chlorophyll was separated from other pigments by using 10 mM KOH as described previously by Li et al. [67]. In a separator funnel, 100 mL of petroleum ether was added to the raw extract in order to obtain two phases: the lower of bright green color, containing chlorophyll, and the upper of yellow-orange color, containing carotenoids. The upper phase was recovered, dried in a Rotavapor and dissolved in 95% ethanol. The concentrations of individual carotenoids were assessed by reverse-phase high-pressure liquid chromatography according to Van Heukelem and Thomas [68]. The composition of CE and TLE is reported in Appendix A, respectively. The CE from *P. tricornutum* contains 97% carotenoids (of which 27% is fucoxanthin) and 3% chlorophylls (Appendix A). TLE is a mixture of lipids (91.7%) and lipophilic pigments including carotenoids (2.3%, of which 72% is fucoxanthin) and chlorophylls (6%, mainly pheophytin) (Appendix A). Both CE and TLE were stored in ethanol at −20 °C under a nitrogen atmosphere and protected from light.

PAL was purchased from Sigma (St. Louis, MO, USA) and dissolved extemporaneously in ethanol at a final concentration of 50 mM. The purity of the PAL used in this study was more than 98%. The PAL solution was filtered and diluted in fetal bovine serum-free Dulbecco’s Modified Eagle’s Medium (DMEM) containing 5% of bovine serum albumin (BSA) and 1% of penicillin and streptomycin mix to reach a 2 mM concentration. The PAL solution was sonicated for 3 min, heated for 10 min at 55 °C and further diluted to reach a 250 µM concentration. The BSA-enriched (5%) DMEM facilitated PAL incorporation into the HepG2 cells, and the PAL/BSA ratio was 3:1. BSA-bound PAL solution was warmed up at 37 °C to ensure BSA binding before treatment.

### 4.2. Cell Culture

HepG2 human liver cells, purchased from the American Type Culture Collection (Manassas, VA, USA), were seeded in enriched DMEM (Sigma, St. Louis, MO, USA) containing 4.5 g/L of glucose, 10% fetal bovine serum, L-glutamine (2 mM) and 1% of penicillin (10,000 IU/mL) and streptomycin mix (10 mg/mL). The cells were kept in a humidified atmosphere of 5% CO_2_. The HepG2 cells were grown until reaching 70–80% confluence before cell treatment, as described below.

### 4.3. Cell Treatment

Human HepG2 cells were seeded at 1.10^2^ cells/cm^2^ and maintained in a culture medium. One day later, the HepG2 cells reached 70% confluence and were washed with phosphate buffered saline and treated with BSA-bound PAL solution at 250 µM, BSA-bound PAL solution and 6 µg/mL of CE (PAL + CE, corresponding to 120 µg/mL of biomass) or 100 µg/mL of TLE (PAL + TLE, equivalent to 600 µg/mL of biomass) for 24 h. In parallel, HepG2 cells were treated with culture medium alone (CTRL) or associated with TLE (CTRL + TLE, 100 µg/mL) or CE (CTRL + CE, 6 µg/mL) for 24 h. The CE and TLE were preserved in ethanol, and the CTRL cells received the same percentage of ethanol as the PAL + TLE cells (i.e., 1.16% ethanol), the experimental condition with the highest ethanol concentration.

### 4.4. Cell Viability and Mortality Assays

The cell viability was assessed by using the MTT assay. The cells were seeded in 96-well culture plates in a volume of 100 μL at a density of 4000 cells/well. After 24 h, the cells were incubated with different concentrations of CE and TLE from *P. tricornutum* (6 to 100 µg/mL) in a serum-free medium. After 24 h of incubation, 50 μL of MTT (Sigma, St. Louis, MO, USA) was added to each well at a concentration of 2.5 mg/mL for 4 h at 37 °C. The cells were washed and lysed with DMSO at 200 μL/well. The absorbance was measured spectrophotometrically at 540 nm. In addition, cell necrosis was determined with the LDH release assay, using a commercial enzymatic kit from Roche (Basel, Switzerland).

### 4.5. Oil Red Staining

HepG2 cells were seeded at a density of 1000 cells/well in 6-well plates, and after 24 h of incubation, they were treated with PAL associated or not with microalgal extracts. After 24 h of treatment, lipid droplets in HepG2 cells were revealed with an Oil Red staining kit from ScienCell Research Laboratories (Carlsbad, CA, USA) according to the manufacturer’s protocols. The images were acquired by optical microscopy using ×400 magnification. Then, the lipid droplets previously stained with Oil Red were solubilized with 2 mL of isopropanol/well. The red coloration of the solution obtained was proportional to the lipid droplet accumulation in the HepG2 cells and was measured spectrophotometrically at 520 nm.

### 4.6. Triglyceride, Cholesterol and Cholesterol Ester Contents in HepG2 Cells

To determine the cellular levels of total and free cholesterol, HepG2 cells were seeded at a density of 1000 cells/well in 6-well plates, while 5000 cells/dish (size 92 × 17 mm) were seeded to measure cellular triglycerides. After 24 h, HepG2 cells were treated with PAL, alone or in combination with microalgal extracts for another 24 h. The cellular levels of total cholesterol and free cholesterol were measured by enzymatic methods using commercial enzymatic kits (Abnova, Taipei, Taiwan), and cellular triglycerides were quantified using a commercial enzymatic kit from Cohesion Biosciences (London, England). For each sample, the cell pellet was solubilized in 1 mL of 0.01 N NaOH at 37 °C for 3 h in order to determine the protein concentrations by the Bio-Rad protein assay (Bio-Rad, Marnes-la-Coquette, France). The cellular levels of cholesterol esters were calculated from the difference between the total cholesterol and free cholesterol values.

### 4.7. RNA Extraction and Real-Time Quantitative Polymerase Chain Reaction (RT-qPCR)

HepG2 cells were seeded at a density of 1000 cells/well in 6-well plates. After 24 h, the cells were treated with PAL (250 µM) associated or not with microalga extracts (6 µg/mL of CE or 100 µg/mL of TLE) for another 24 h at 37 °C. The cells were lysed with TRIzol Reagent (Thermo Fisher, Waltham, MA, USA) and total RNA was isolated following the manufacturer’s instructions. The mRNAs (1 μg) were then reverse-transcribed into cDNA using iScript Reverse Transcriptase (Bio-Rad) according to the manufacturer’s instructions. An initial priming step of 5 min at 25 °C was followed by reserve transcription for 30 min at 42 °C and a reverse-transcriptase inactivation for 5 min at 85 °C. Quantitative PCR was performed on a MyiQ2 Real-Time PCR Detection System (Bio-Rad) using iQ™ SYBR Green Supermix. PCR was carried out for 45 cycles of 95 °C for 30 s and 60 °C for 30 s. The relative expression levels of the studied genes were standardized to the *β-actin* (*ACTB*) reference gene expression—except that of *LXR/NR1H3*, which was normalized to that of the *18S* reference gene—using the ΔΔCT method. The sequences of the primers used are listed in Appendix A.

### 4.8. Data Analysis

Results were obtained from triplicate experiments, and the data presented are the means +/− standard deviations (SDs) obtained after at least three independent experiments. The analysis of variance by one-way ANOVA followed by a Fisher’s least significant difference post hoc test (LSD) was performed. Statistical analyses were carried out with Statgraphics Plus 5.1 software (Manugistics Inc., Rockville, MD, USA), and the results were considered statistically different for values of *p <* 0.05, *p <* 0.01 or *p <* 0.001 as indicated in the figure legends.

## 5. Conclusions

This study highlights the preventive effect of total lipophilic and carotenoid extracts of the marine microalga *P. tricornutum* on the hepatic accumulation of triglycerides and cholesterol. Moreover, only the carotenoid extract reduced the accumulation of lipid droplets in PAL-treated HepG2 cells. This observation could be explained by an additional inhibitory effect of CE on lipolysis in parallel with the stimulation of the de novo lipogenesis pathway, whereas TLE acts only on the latter. This suggests a greater protective effect of carotenoids against NAFLD compared to the total lipophilic compounds of TLE. Therefore, carotenoid extracts of *P. tricornutum* might become a promising nutritional ingredient for a therapeutic strategy against hepatic disorders such as NAFLD associated with metabolic syndrome and obesity. Finally, by comparing CE’s and TLE’s effects, this study also highlights the complex interactions between the bioactive molecules contained in TLE such as fucoxanthin, chlorophylls and *n*-3 LC-PUFA. Further studies should be conducted with a purified lipid extract, free of chlorophylls and carotenoids, in order to better elucidate the specific effects of lipids and carotenoids of *P. tricornutum*.

## Figures and Tables

**Figure 1 molecules-25-02845-f001:**
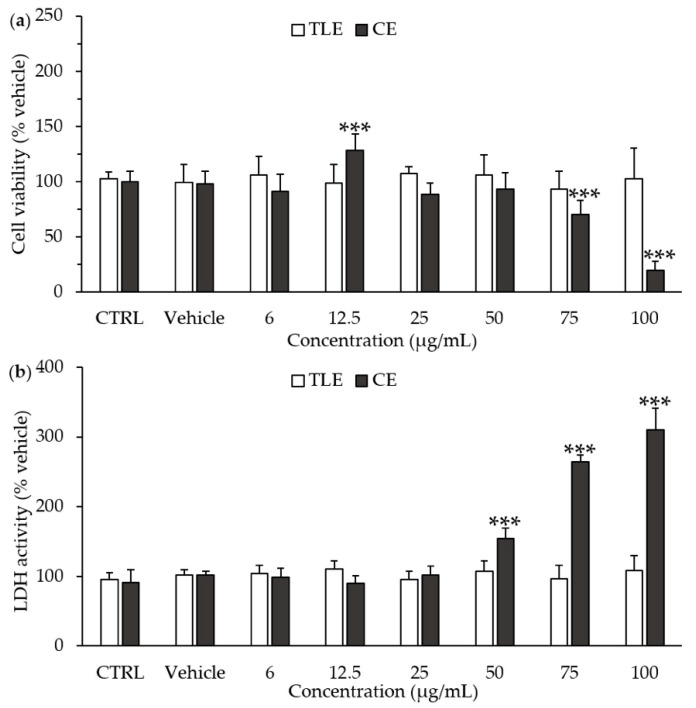
Effect of total lipophilic and carotenoid extracts from *P. tricornutum* on the cell viability and necrosis of HepG2 cells. The HepG2 cells were treated with the culture medium (control, CTRL), the vehicle (0.66% ethanol) or different concentrations of total lipophilic extract (TLE) and carotenoid extract (CE) from *P. tricornutum* (6–100 µg/mL) for 24 h, and cytotoxicity assays were performed. (**a**) Cell viability was evaluated by the MTT assay; (**b**) Necrosis of HepG2 cells was estimated by the measurement of extracellular lactate dehydrogenase (LDH) activity. Data are the means ± standard deviations from at least three independent experiments. Statistical significance was determined using ANOVA with post-hoc Fisher’s test and (an) asterisk(s) indicate significant difference compared to the vehicle with *** *p* < 0.001.

**Figure 2 molecules-25-02845-f002:**
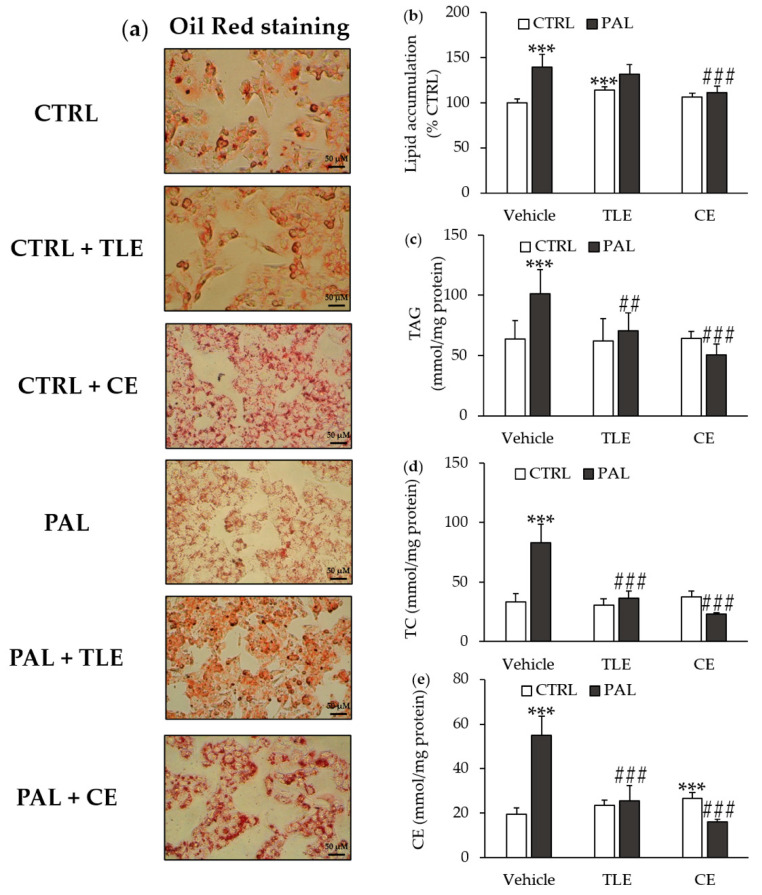
Effect of total lipophilic and carotenoid extracts from *P. tricornutum* on non-alcoholic fatty liver disease (NAFLD). The HepG2 cells were treated with the control vehicle (CTRL, 1.16% ethanol) alone or associated with total lipophilic extract (CTRL + TLE, 100 µg/mL) or carotenoid extract (CTRL + CE, 6 µg/mL) for 24 h. In parallel, the HepG2 cells were exposed to palmitate vehicle (PAL, 250 µM) alone or associated with the TLE (PAL +TLE) or the CE (PAL + CE) for 24 h. (**a**) Lipid droplets were detected by Oil Red staining, original magnification × 400; (**b**) Lipid accumulation was quantified by the solubilization of lipid droplets stained in Oil Red with isopropanol; (**c**) Intracellular triglyceride (TAG) values; (**d**) Intracellular total cholesterol (TC) values; (**e**) Intracellular cholesterol ester (CE) values. Data are the means ± standard deviations from at least three independent experiments. Statistical significance was determined using ANOVA with post-hoc Fisher‘s test, and asterisks indicate significant differences compared to the CTRL vehicle with *** *p* < 0.001. Hashtags indicate significant differences compared to the PAL vehicle with ## *p* < 0.01 or ### *p* < 0.001.

**Figure 3 molecules-25-02845-f003:**
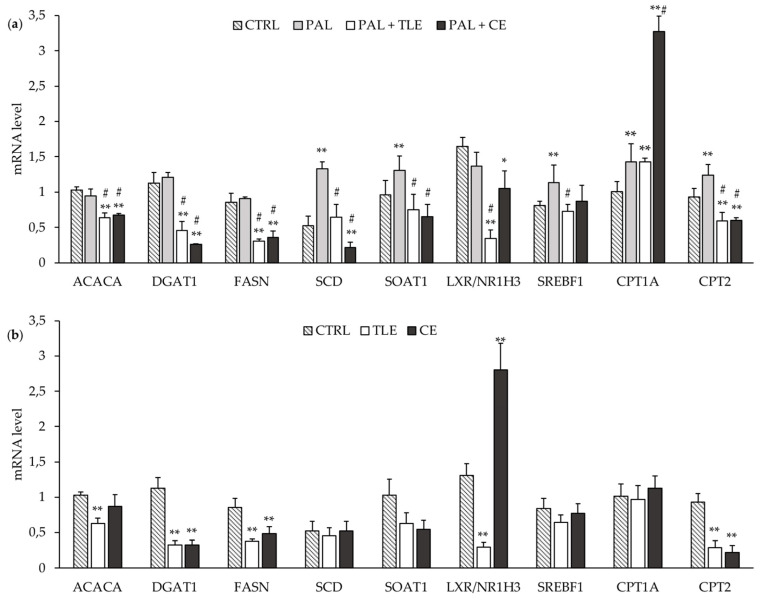
Impact of total lipophilic and carotenoid extracts from *P. tricornutum* on mRNA levels of several genes involved in lipid metabolism regulation in palmitate (PAL)-treated HepG2 cells (**a**) and control (CTRL) cells (**b**). The HepG2 cells were treated with control medium and 1.16% ethanol (CTRL), PAL (250 µM) alone or associated with total lipophilic extract (PAL + TLE, 100 µg/mL) or carotenoid extract (PAL + CE, 6 µg/mL) for 24 h before RNA extraction. mRNA levels of genes were quantified using RT-qPCR. Data are the means ± standard deviations from at least three independent experiments. Statistical significance was determined using ANOVA with post-hoc Fisher’s test, and asterisk(s) indicate significant differences compared to the CTRL vehicle with * *p* < 0.05 and ** *p* < 0.001. Hashtags indicate significant differences compared to the PAL vehicle with # *p* < 0.001.

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
