# Peer review of "Effect of Carotenoids from Phaeodactylum tricornutum on Palmitate-Treated HepG2 Cells"

_molecules, 2020, doi:10.3390/molecules25122845_

Round 1

Reviewer 1 Report

The paper “Effect of carotenoids from Phaeodactylum 3 tricornutum on palmitate-treated HepG2 cells” focuses on the effect of microalgae extract on an in vitro model of Non-alcoholic fatty liver disease induced by palmitate treatment.

The topic is interesting and fully falls into the scope of the Journal. The paper is well composed, with clear results and exhaustive discussion.

Here are only some minor points which should be addressed:

-Line 83: some more details about P. tricornutum should be provided (i.e. habitat, diffusion in the world)

-Line 96: you can delete the last sentence of the Introduction as it better deals with  Results.

-Though the acronyms have been explained in the Abstract, you should explain them the first time they are mentioned in the Introduction (i.e. CE and TLC, which are actually reported in the last sentence)

Line 274: which is the final concentration of ethanol? Different values are reported ( legend of figure 1 and line 295), please specify.

Concluding remarks should contain less data, more explanation about the biological meaning of the whole investigation , impact on the current knowledge about therapeutic strategies against Non-alcoholic fatty liver disease

After these minor changes, the paper can be accepted for publication.

Reviewer 2 Report

This very interesting paper describes the effects of Phaeodactylum tricornutum (a marine microalgae) extracts on HeG2 cells, a cellular model of non-alcoholic fatty liver disease. These cells were exposed to palmitate in order to induce lipotoxicity and hepatic steatosis, and treated with carotenoid and a total lipophilic extracts from P. tricornutum. Cytotoxicity assays were performed, as well as, the assessment of lipid droplets accumulation, cellular levels of triglycerides, total cholesterol and cholesterol esters, and the impact of extracts from on mRNA levels of lipid metabolism and β-oxidation related genes.

The manuscript falls in the scope of Molecules, the research was very well conducted and the paper is clearly written.

In my opinion, the manuscript should be accepted after minor revisions, such as:

  • Minor English spell checking;
  • Avoid to use abbreviations in the abstract;
  • A list of acronyms and abbreviations is advised;
  • There is no need to describe p values in the text (eg… lines 106 and 122, etc.) since they are indicated in figures legends; in this context, quality of Figure 3 should be increased.
  • Line 218: “According to the present study, previous studies…” – I think it should be: “in accordance with…” or “Corroborating the present study…”
  • Lines 346 – 327 – This sentence should be changed since P. tricornutum is not a functional food.
